# Treatment and Prognosis of COVID-19 Associated Olfactory and Gustatory Dysfunctions

**DOI:** 10.3390/jpm11101037

**Published:** 2021-10-16

**Authors:** Min Young Seo, Seung Hoon Lee

**Affiliations:** Division of Rhinology, Department of Otorhinolaryngology-Head and Neck Surgery, Korea University Ansan Hospital, Korea University College of Medicine, Ansan 15355, Korea; chariseoma@gmail.com

**Keywords:** COVID-19, olfactory dysfunction, gustatory dysfunction, clinical course, treatment

## Abstract

Olfactory and gustatory dysfunctions are important initial symptoms of coronavirus disease 2019 (COVID-19). However, the treatment modality for these conditions has yet to be clearly established. Therefore, most physicians have been administering empirical treatments for COVID-19-associated olfactory dysfunction, including topical or systemic steroid supplementation and olfactory training. In this literature review, we summarize the clinical course and effects of various treatments currently being conducted in patients with COVID-19-associated olfactory and gustatory dysfunctions.

## 1. Introduction

Following the reports of several severe pneumonia cases due to severe acute respiratory syndrome coronavirus 2 (SARS-CoV-2) in December 2019 in Wuhan, China, the virus has spread worldwide as a global pandemic, earning the name coronavirus disease 2019 (COVID-19). Currently, the disease has a reported morbidity of more than 220 million patients and a mortality of more than 4.6 million [1]. In affected patients, the symptoms of COVID-19 can vary from an asymptomatic presentation to acute respiratory disease syndrome (ARDS) and even death. Moreover, despite a high viral load, many patients do not present any symptoms during the initial stages of this disease, thus contributing to the difficulty involved in COVID-19 control measures. In the early days of the pandemic, respiratory symptoms, including cough, runny nose, sputum, shortness of breath, and systemic symptoms such as fever and weakness, were reported as its main presenting symptoms [2,3]. Interestingly, subsequent studies reported that the loss of smell and taste was an important clinical and pathognomonic feature of COVID-19 [4,5]. Despite this discovery, the treatment modality for olfactory and gustatory dysfunctions associated with COVID-19 has yet to be clearly established.

Post-infectious olfactory dysfunction is a disease which occurs after upper airway viral infection and is characterized by olfactory dysfunction persisting even after the resolution of other symptoms associated with upper respiratory infection. Last year, we published an article about the trend of olfactory dysfunction in mild COVID-19 patients. In this study, we reported that COVID-19 infection-associated olfactory dysfunction was regarded as having a sensory neural cause, characterized by a quantitative disorder (reduced or absence of olfaction), such as post-infectious olfactory dysfunction [6]. In the case of post-infectious olfactory dysfunction, the optimal treatment strategies remain unclear. A recently published meta-analysis by Hura et al. reported that based on the evidence, olfactory training is a recommendation for the treatment of post-infectious olfactory dysfunction and the application of topical or systemic steroids is an option in select patients [7]. As a result, most physicians administer empirical treatments for the management of COVID-19-associated olfactory dysfunction, including topical or systemic steroid supplementation and olfactory training. Topical or systemic steroid application improves olfactory dysfunction by exerting anti-inflammatory effects and modulating the function of olfactory receptor neurons [8]; however, it was reported that the effect of steroid supplementation is minimal on post-infectious olfactory dysfunction, since the improvement of olfactory dysfunction only occurred when treatment was initiated in the acute phase of an infection [9]. Olfactory training, on the other hand, is one of the most established treatments for olfactory dysfunction, which has been reported to significantly improve olfactory function in patients with post-infectious olfactory dysfunction [10,11]. In this review, we summarized the clinical course and effects of various treatments currently being conducted in patients with COVID-19-associated olfactory and gustatory dysfunctions.

### 1.1. Pathogenesis

Various studies have reported on the pathogenesis of COVID-19 infection-associated olfactory dysfunction. Kandemiril et al. published an article about olfactory bulb magnetic resonance imaging (MRI) findings in persistent COVID-19 anosmia patients. They reported that olfactory cleft opacification was observed in 73.9% of the patients, and 43.5% and 60.9% of the patients had below-normal olfactory bulbs and shallow olfactory sulci, respectively. In addition, 54.2% of the patients exhibited a change in the normal shape (inverted J shape) of the olfactory bulb, and 91.3% of the patients exhibited abnormal signal intensity of the olfactory bulb [12]. Moreover, Chiu et al. reported a case of olfactory bulb atrophy after 2 months of COVID-19 infection-associated olfactory dysfunction, compared with pre-COVID imaging [13]. Politi et al. also reported a case involving the serial evaluation of the olfactory bulb using MRI during COVID-19 infection. They reported that olfactory bulbs were thinner and slightly less hyperintense at 28 days after the onset of symptoms. Therefore, the authors suggested that direct or indirect injury to the olfactory neuronal pathway lead to olfactory dysfunction in COVID-19 infection [14].

According to a study involving inflammatory cytokine evaluations in the olfactory epithelium of the deceased patients due to COVID-19 infection by Torabi et al., they reported that level of TNF-α was significantly increased in the COVID-19 group than control group [15]. Therefore, they suggested that direct inflammation of the olfactory epithelium can lead to sensorineural olfactory dysfunction in COVID-19. Damage to the olfactory epithelium is also confirmed by other studies. Li et al. reported that SARS-CoV-2 infects the ciliated cells in the human nasal epithelium and causes deciliation. They suggested that this defect in olfactory cilia leads to olfactory loss after COVID-19 infection [16]. Vaira et al. also reported significant deterioration of olfactory epithelium using histopathological evaluations of the olfactory epithelia of COVID-19 patients who presented with anosmia of more than 3 months in duration. In this study, the subjects also underwent contrast-enhanced MRI of the nasal cavity and brain, and showed no abnormalities in the volume of the olfactory bulb and cleft and no signal abnormalities [17]. Therefore, they suggested that disruption and desquamation of the olfactory epithelium is the underlying mechanism in COVID-19-related olfactory dysfunction. These findings have important implications when considering treatment for olfactory dysfunction in COVID-19 patients; it should be considered that the olfactory epithelium should be the target.

Recently, an interesting article was published, associating COVID-19-related anosmia with viral persistence in the human olfactory epithelium. They reported that in patients with long-lasting/relapsing olfactory dysfunction after COVID-19 infection, SARS-CoV-2 RNA was detected in cytological samples from olfactory mucosa, but not in nasopharyngeal samples. Therefore, they reported that the persistence of COVID-19-associated olfactory dysfunction is linked to the inflammation caused by persistent infection [18]. In addition, they also suggested that persistent olfactory dysfunction might result from direct damage to the olfactory sensory neurons of the olfactory epithelium and retrograde neuro-invasion of SARS-CoV-2 through induced neuro-inflammation in the olfactory route.

### 1.2. Prevalence

According to a recently published meta-analysis by Saniasiaya et al. [19], it was reported that among 27,492 COVID-19 patients, 47.85% of patients complained of olfactory dysfunction in olfactory evaluations using subjective olfactory assessments. Moreover, according to subgroup analysis by racial differences, subjective olfactory dysfunction was observed in 54.40% of Europeans, 51.11% of North Americans, 31.39% of Asians, and 10.71% of Australian COVID-19 patients. However, upon olfactory evaluation using objective olfactory assessments (psychophysical test), 72.1% of patients were confirmed to have olfactory dysfunction. Similarly, various studies have reported that approximately 43.93–56.4% of COVID-19 patients complained of taste disturbances [20]. In particular, our authors published an article on 62 mild COVID-19 patients who were isolated in the Gyeonggi International Living and Treatment Support Center (LTSC) during the early stage of the pandemic (May 2020), reporting that 24.2% and 17.7% complained about subjective olfactory and gustatory dysfunction, respectively. Furthermore, when we performed an objective olfactory function evaluation using the Cross-Cultural Smell Identification Test (CC-SIT), we confirmed that all patients with reported olfactory dysfunction had a reduced sense of smell [6].

### 1.3. Prognosis

Several studies have reported the clinical course of COVID-19-associated olfactory and gustatory dysfunctions [21,22,23,24,25,26]. According to these results, approximately 8.57–25% of these patients did not have improved subjective olfactory function at 1–2 months after symptom onset, and 11.2% did not have improved subjective olfactory dysfunction at 6 months after symptom onset. On the other hand, 33.3% and 58.4% of patients showed complete recovery from olfactory dysfunction within 1 month and at 6 months after symptom onset, respectively. In addition, in a cross-sectional study of over 700 healthcare workers, a reduced sense of smell was still reported in 52% of patients even 3 to 7 months after the onset of symptoms [27]. Using an objective assessment of olfactory function recovery, these studies showed that 15.3% and 5% of patients did not recover to normal olfactory function at 2 and 6 months after symptom onset, respectively. Furthermore, a recent article by our authors, including 53 patients who recovered from COVID-19, showed that among 38 patients who experienced olfactory dysfunction, 92.1% of them reported the subjective normalization of olfactory function, although only 52.6% of them were confirmed to have normal olfactory function via CC-SIT score evaluation at 3 months after symptom onset [25]. In addition, according to a prospective study of 183 adult COVID-19 patients, 110 patients complained of the sudden onset of olfactory dysfunction subjectively. Among the 110 patients, 85 (77.3%) patients reported complete recovery, 22 (20.0%) patients reported partial improvement, and three (2.7%) patients reported that their olfactory function was not improved or worse 6 months after symptom onset. They also reported the olfactory outcomes using psychophysical olfactory evaluation using University of Pennsylvania Smell Identification Test (UPSIT). In this study, 145 underwent the UPSIT at 6 months after symptom onset; among the 145 patients, 112 patients reported that their olfactory function was normal. However, in these patients, 46 (41.1%), 11(9.8%) and three (2.3%) patients showed confirmed mild, moderate and severe olfactory dysfunction, respectively. Furthermore, six (5.4%) patients confirmed anosmia according to their UPSIT scores [26]. According to another study by Bussiere et al., 19.5% of patients still had objectively confirmed olfactory impairments even after 3 to 7 months after the onset of symptoms [27]. Therefore, we observe that the prevalence of olfactory dysfunction might be underestimated due to the majority of studies about the prognosis of olfactory dysfunction in COVID-19 being based on subjective evaluations of olfactory function. Moreover, we also have confirmed that olfactory dysfunction could persist for a long time in a number of patients. Therefore, the confirmation of olfactory dysfunction in the early stage using an objective modality and the immediate initiation of proper management in COVID-19 patients with olfactory dysfunction is necessary.

Regarding gustatory dysfunction, another article by our authors showed that 73.6% of patients reported complete subjective recovery at 6 months after onset, whereas 12% of them reported no improvement in gustatory dysfunction [24].

### 1.4. Treatment

As mentioned previously, there have been no validated treatments for COVID-19-associated olfactory and gustatory dysfunctions. As such, most physicians have opted to administer empirical treatments using topical or systemic steroids and olfactory training. Therefore, we have attempted to summarize the olfactory outcomes of each treatment modality as follows: (1) intranasal topical steroid vs. control; (2) systemic steroid supplementation vs. control; (3) intranasal topical steroid with olfactory training vs. olfactory training alone; (4) systemic steroid supplementation with olfactory training vs. olfactory training alone; and (5) olfactory training alone. In addition, we have summarized the results of gustatory dysfunction treatment using triamcinolone paste.

## 2. Treatment of Olfactory Dysfunction

### 2.1. Intranasal Topical Steroid vs. Control

In an investigator-initiated, randomized, double blind, parallel-arm, placebo-controlled clinical trial of 276 patients (case (betamethasone 0.1 mg/mL, *n* = 138) vs. control (0.9% NaCl solution, *n* = 138)) conducted by Rashid et al. [28], it was reported that three drops of betamethasone taken thrice daily did not facilitate a reduction in the recovery time from acute anosmia. In contrast, another randomized controlled study including 120 patients (case (fluticasone nasal spray with nasal saline irrigation, *n* = 60) vs. control (nasal saline irrigation, *n* = 60)) conducted by Singh et al. [29] reported that five days of fluticasone nasal spray treatment significantly improved olfactory function, as compared with the control group. However, it should be noted and taken into consideration that these two studies did not perform a validated olfactory assessment. The results of these studies are presented in Table 1.

### 2.2. Systemic Steroid Supplementation vs. Control

According to a randomized case-control study of 18 patients (case (systemic prednisone 1 mg/kg/day and nasal irrigation with betamethasone, ambroxol, and rinazine, *n* = 9) vs. control (untreated, *n* = 9)) conducted by Vaira et al. [30], 15 days of systemic steroid administration significantly reduced the prevalence of anosmia and hyposmia after 20 and 40 days of symptom onset, respectively. Furthermore, an objective olfactory function assessment using the Connecticut Chemosensory Clinical Research Center (CCCRC) score showed a significant improvement in the treatment group compared with the no-treatment group. Therefore, it was concluded that a therapeutic modality of drug polytherapy, including steroids, reduces the prevalence of olfactory dysfunction.

### 2.3. Intranasal Topical Steroid with Olfactory Training vs. Olfactory Training Alone

Several studies have compared whether the addition of topical intranasal steroids with initial olfactory training affects the improvement of olfactory function in patients with COVID-19-associated olfactory dysfunction. A randomized controlled trial of 100 patients (case (mometasone furoate 2 puffs (100 μg) once daily with olfactory training, *n* = 50) vs. control (olfactory training only, *n* = 50)) conducted by Abdelalim et al. [8] reported that both groups showed significantly improved subjective olfactory function after three weeks of treatment, showing no significant differences between the two groups. Similarly, the recovery rate was not significantly different between the two (62% vs. 52%, *p* = 0.31). Therefore, they concluded that intranasal spray offers no superiority over olfactory training. Furthermore, according to a prospective, double-blinded, randomized clinical trial including 77 patients (case (mometasone furoate 2 puff (100 μg) once daily with olfactory training, *n* = 39) vs. control (0.9% NaCl solution with olfactory training, *n* = 38)) conducted by Kasiri et al. [31], subjective olfactory function was reported to have significantly improved in the treatment group, but objective olfactory function assessment using the University of Pennsylvania smell identification test (UPSIT) did not show significant differences between the two groups after four weeks of treatment. Despite this, they also reported that the recovery rate to normal olfactory function was significantly higher in the treatment group. Thus, they concluded that the combination of intranasal steroids with olfactory training resulted in a greater improvement of symptoms among patients with COVID-19-associated olfactory dysfunction. The results of these studies are presented in Table 2.

### 2.4. Systemic Steroid Supplementation with Olfactory Training vs. Olfactory Training Alone

Le Bon et al. [32] performed a prospective case-control study (case (methylprednisolone 32 mg for 10 days with olfactory training, *n* = 9) vs. control (olfactory training alone, *n* = 18)), reporting that only the case group showed an improvement in their olfactory score, which was measured according to the threshold discrimination identification (TDI) score using the Sniffin’ Sticks test. Moreover, Saussez et al. [33] performed a case-control study (case 1 (methylprednisolone 0.5 mg/kg/day for 10 days with olfactory training, *n* = 59) vs. case 2 (mometasone furoate 2 puff [100 μg] once daily with olfactory training, *n* = 22) vs. control (olfactory training alone, *n* = 71)) and reported that the TDI score was significantly improved after treatment in all groups, with the systemic steroid supplement group showing the highest degree of improvement at one month of treatment. However, this superiority did not remain at two months after treatment, since the degree of olfactory improvement in the other groups became similar to that of the systemic steroid supplement group. Therefore, with consideration of the risk–benefit ratio, it was concluded that the benefit of systemic steroid supplementation could not be demonstrated. The results of these studies are presented in Table 3.

### 2.5. Olfactory Training Alone

Our authors could find only observational studies that conducted olfactory training alone for patients with COVID-19-associated olfactory dysfunction. Denis et al. [34] reported that after performing olfactory training and visual stimulation for an average of four weeks in 548 patients, the recovery rate (VAS score increase of 2 or more) was found to be 73.3% in the group of patients who trained for more than 28 days and 59% in the group who trained for less than 28 days. As such, they suggested that olfactory training for more than 28 days was important for the management of olfactory dysfunction. In another study, recently published by our authors, regarding the effect of olfactory training in patients with olfactory dysfunction even after three months of symptom onset, 10 patients underwent olfactory training for 8 weeks using four Korean odorants: pine, peppermint, cinnamon, and lemon. It was found that the objective olfactory function score using the Cross-Cultural Smell Identification Test (CC-SIT) was significantly increased after olfactory training, with 70% of patients reporting a recovery to normal olfactory function [25].

## 3. Treatment of Gustatory Dysfunction

Regarding treatment for COVID-19-associated gustatory dysfunction, we found only one randomized controlled study of 120 patients (case (triamcinolone paste with normal saline gargle, *n* = 60) vs. control (normal saline gargle, *n* = 60)), which was conducted by Singh et al. [29]. They reported that five days of treatment using an oral application of triamcinolone paste significantly improved gustatory function (bitter, sweet, salty, and sour) in the case group, as compared with the control group.

## 4. Conclusions

To date, there has been no validated treatment for COVID-19-associated olfactory and gustatory dysfunctions. For the treatment of olfactory dysfunction after COVID-19 infection, we should consider the pathogenesis of the disease. As mentioned above, destruction of the olfactory epithelium and neuro-inflammation of the olfactory bulb due to direct invasion of the virus via the olfactory pathway are considered to be the main pathogenesis of olfactory dysfunction in COVID-19 patients. Based on these results, it is observed that COVID-19 infection is not a conductive but a sensorineural cause of olfactory dysfunction. These clinical features are similar to those of post-infectious olfactory dysfunction caused by other types of viral infection. Therefore, physicians should consider this when determining the treatment policy. Considering the risks and benefits of conducting olfactory training, we believe that olfactory training has its advantages in the management of COVID-19 associated olfactory dysfunction. Moreover, the results of the Delphi process, which involved the opinions of experts from the Clinical Olfaction Working Group, reported that 95% of experts agreed that olfactory training should be prescribed as soon as possible in cases of COVID-19-associated olfactory dysfunction [35]. In contrast, the results of topical and oral steroid therapies remain inconsistent, and opinions on their use are still divided. Thus, we believe that further well-designed studies will be helpful to clarify whether topical and oral steroid therapies are applied in consideration of the risks and benefits of the clinical situation. In addition, physicians should also consider persistent infection in patients with long-lasting or relapsing olfactory dysfunction. Although there is no clinical evidences for this yet, we suggest that further clinical research from a theoretical perspective, based on the virologic, molecular, and cellular studies, may be of great help in understanding the characteristics of this disease.

## Figures and Tables

**Table 1 jpm-11-01037-t001:** Assessment of olfactory improvement according to intranasal steroid application only.

Study ID ^Reference^	Study Design	Number of Patients	Olfactory Assessment	Intervention	Results
Rashid et al. [28]	Randomized controlled trial	276 (138 vs.138)	Self-reported time of recovery from anosmia	Betamethasone 0.1 mg/mL	Betamethasone did not facilitate the recovery time
Singh et al. [29]	Randomized controlled trial	120 (60 vs. 60)	Non-validated assessment using 5 odorants (musky, pungent, camphoraceous, floral, peppermint)	Fluticasone nasal spray 2 puff once a day for 5 days	Fluticasone significantly improved olfactory function

**Table 2 jpm-11-01037-t002:** Assessment of olfactory improvement according to additional intranasal steroid application with olfactory training.

Study ID ^Reference^	Study Design	Number of Patients	Olfactory Assessment	Intervention and Olfactory Training Regimen	Results
Abdelalim et al. [8]	Randomized controlled trial	100 (50 vs.50)	VAS (0–10)	Mometasone furoate (100 µg) for 3 weeks; OT regimen: rose, lemon, and clove	Mometasone furoate had no superior benefits in subjectivity score and recovery rate over olfactory training
Kasiri et al. [31]	Randomized controlled trial	77 (39 vs. 38)	VAS (0–10), UPSIT	Mometasone furoate (100 µg) for 4 weeks; OT regimen: rose, lemon, clove, and eucalyptus	Mometasone furoate significantly improved subjective score and recovery rate

VAS, visual analogue scale; OT, olfactory training; UPSIT, University of Pennsylvania Smell Identification Test.

**Table 3 jpm-11-01037-t003:** Assessment of olfactory improvement according to additional systemic steroid application with olfactory training. TDI, threshold discrimination identification; OT, olfactory training.

Study ID ^Reference^	Study Design	Number of Patients	Olfactory Assessment	Intervention and Olfactory Training Regimen	Results
Le Bon et al. [32]	Prospective case-control study	41 (32 vs. 9)	TDI score using the Sniffin’Sticks test	Methylprednisolone 32 mg for 10 days; OT regimen: rose, lemon, clove, and eucalyptus	Systemic steroid supplement significantly improved TDI score
Saussez et al. [33]	Prospective case-control study	152 (59 vs. 22 vs. 71)	TDI score using the Sniffin’Sticks test	methylprednisolone 0.5 mg/kg/day for 10 days; Mometasone furoate (100 µg) for 1 month; OT regimen: coffee, perfume, essential oils	The benefit of steroid treatment cannot be demonstrated

TDI, threshold discrimination identification; OT, olfactory training.

## Data Availability

Not applicable.

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
