# Peer review of "Treatment and Prognosis of COVID-19 Associated Olfactory and Gustatory Dysfunctions"

_jpm, 2021, doi:10.3390/jpm11101037_

Round 1

Reviewer 1 Report

I reviewed this interesting review of the literature which summarizes the results obtained so far in the treatment of chemosensory disorders in Covid-19 patients. The topic is very current and, given the high prevalence of persistent disorders in COVID-19 patients, this will represent a difficult challenge that awaits smell specialists in the near future. To the best of my knowledge, no similar reviews have yet been published.

I have some observations that I would like the authors to clarify:

1. the paper is organized as a contemporary review but this type of manuscript is not foreseen by the journal. According to the journal's guidelines, authors should follow the PRISMA guidelines for reviews. Authors should then report how the research of the studies was performed and enter the flowchart required by the PRISMA guidelines

2. in the introduction I would add a paragraph listing the possible therapies proposed up to now for post-viral olfactory disorders. It is useful for the reader to get a general picture of the topic and of what was proposed before COVID-19

3. line 50-51. please double check the sentence, it is not clear what 10% of COVID-19 patients are referring to

4. paragraph 1.3 (prognosis) is fundamental to understand the importance of the topic that the authors deal with. the authors should emphasize more that the problem that COVIDwl-19 is posing is that a large number of patients have long-term disorders and for this reason it is important to seek a therapy that prevents and / or treats the onset of these disorders. There are several six-month follow-up studies that can be cited by the authors

5. a final point that I would deal with in the introduction or discussion is that related to the possible pathogenesis of persistent olfactory disturbances: persistence of the virus or of viral fragments in the olfactory epithelium? (Dias de Melo et al 10.1126/scitranslmed.abf8396) complete olfactory epithelium disruption? (vaira et al doi: 10.1017/S0022215120002455) local immune factors? (Saussez et al. doi: 10.1111/ene.14994)

Author Response

                                                                                               Sep 13, 2021

Dear Editor and Reviewers; 

I really appreciate the editor and reviewers of the ‘Journal of Personalized Medicine’ for taking their time to review my article. I have made some corrections and clarifications in the manuscript after going over the reviewer’s comments.

The answers to the comment are summarized in separate pages.

We would be happy if the revised manuscript is more suitable to publication in the ‘Journal of Personalized Medicine’. I thank you again for the constructive review by the editors and referees.

Sincerely yours,

Seung Hoon Lee, MD, PhD

Professor,

Department of Otorhinolaryngology-Head and Neck Surgery, Ansan Hospital, 123, Jeokgeum-ro, Danwon-gu, Ansan-city, Gyeonggi-do, 15355, Korea

Tel: +82-31-412-5270

Fax: +82-31-412-5174

       Re: jpm-1365638

Treatment and prognosis of COVID-19 associated olfactory and gustatory dysfunctions

Referee(s) comments to author: 

Reviewer: 1

Comments and Suggestions for Author

I reviewed this interesting review of the literature which summarizes the results obtained so far in the treatment of chemosensory disorders in Covid-19 patients. The topic is very current and, given the high prevalence of persistent disorders in COVID-19 patients, this will represent a difficult challenge that awaits smell specialists in the near future. To the best of my knowledge, no similar reviews have yet been published.

I have some observations that I would like the authors to clarify:

  1. The paper is organized as a contemporary review but this type of manuscript is not foreseen by the journal. According to the journal's guidelines, authors should follow the PRISMA guidelines for reviews. Authors should then report how the research of the studies was performed and enter the flowchart required by the PRISMA guidelines

Answer: Thank you for your comment. We also found the type of publication and the paragraph as ‘Reviews: These provide concise and precise updates on the latest progress made in a given area of research. Systematic reviews should follow the PRISMA guidelines.’ Therefore, we checked other review articles which published at the ‘Journal of Personalized Medicine’ and found that, several literature review articles also published. So, we interpret it that we should follow the PRISMA guidelines only in case of systematic review, not in other form of review article. Our authors also agree that the results presented through systematic review and meta-analysis suggest more valuable than literature review. Recently, many patients have complained of olfactory dysfunction after COVID-19 infection, but there is no validated treatment for it. Therefore, this study was conducted with the belief that it was necessary to summarize the treatment outcome by a physician who had experience in treating patients. However, for the systematic review of research on this topic, among the various clinical studies, the modality of olfactory outcome evaluation according to treatment was not made uniformly, and the clinical characteristics of the studies subjects showed difference. Therefore, we regarded that literature review is more suitable in the current situation. Because we conducted a literature review, based on the results of recently published several studies, we could only suggest some direction in the treatment of the current situation, but could not present a clear conclusion. As the COVID-19 pandemic continues, the number of patients complaining of olfactory dysfunction will continue to increase, and the demand for treatment will also increase. Therefore, if we summarize the current treatment outcome and share it, I think it will be helpful to many physicians for establish of treatment policy. In addition, if a certain amount of treatment is established and the results of various well-designed studies are published, valuable conclusions using systematic review and meta-analysis can be presented.

  1. 2. In the introduction I would add a paragraph listing the possible therapies proposed up to now for post-viral olfactory disorders. It is useful for the reader to get a general picture of the topic and of what was proposed before COVID-19.

Answer: Thank you and we also agree with your comment. As your recommendation, we added a paragraph as ‘Post-infectious olfactory dysfunction (PIOD) is a disease which occurs after upper airway viral infection and characterized by olfactory dysfunction persist even after the resolution of other symptoms associated with upper respiratory infection. Last year, our authors published an article about trend of olfactory dysfunction in mild COVID-19 patients. In this study, we reported that COVID-19 infection associated olfactory dysfunction was regarded as sensory neural cause, such as PIOD [6]. In case of post-infectious olfactory dysfunction, the optimal treatment strategies still remain un-clear. Recently published meta-analysis by Hura et al. reported that based on the evidence, olfactory training is a recommendation for the treatment of PIOD and the application of topical or systemic steroids is an option in select patients [7].’ in our introduction section.

  1. Line 50-51. please double check the sentence, it is not clear what 10% of COVID-19 patients are referring to

Answer: Thank you for your comment, we made a mistake about it. We insert the missed word ‘Austrailian’. I’m sorry for our mistake.

  1. Paragraph 1.3 (prognosis) is fundamental to understand the importance of the topic that the authors deal with. The authors should emphasize more that the problem that COVID-19 is posing is that a large number of patients have long-term disorders and for this reason it is important to seek a therapy that prevents and / or treats the onset of these disorders. There are several six-month follow-up studies that can be cited by the authors

Answer: Thank you for your comment. We added more contents in our prognosis section as ‘In addition, according to a prospective study of 183 adult COVID-19 patients, 110 patients complained of a sudden onset of olfactory dysfunction subjectively. Among the 110 patients, 85 (77.3%) patients reported complete recovery, 22 (20.0%) patients re-ported partial improvement, and 3 (2.7%) patients reported that their olfactory function was not improved or worse at 6 months after symptom onset. They also reported the olfactory outcomes using psychophysical olfactory evaluation using University of Pennsylvania Smell Identification Test (UPSIT). In this study, 145 underwent the UP-SIT at 6 months of symptom onset, among the 145 patients, 112 patients reported that their olfactory function is normal. However, in these patients, 46 (41.1%), 11(9.8%) and 3 (2.3%) patients confirmed mild, moderate and severe olfactory dysfunction, respectively. Furthermore, 6 (5.4%) patients confirmed anosmia according to UPSIT score [19]. According to another study by Bussiere et al. they reported that 19.5% of patients still had objectively confirmed olfactory impairment even after 3 to 7 months after on-set of symptoms [20]. Therefore, we regarded that the prevalence of olfactory dysfunction might underestimated due to majority of studies about prognosis of olfactory dysfunction in COVID-19 were based on evaluation of subjective olfactory function. Moreover, we also have confirmed that olfactory dysfunction could persist for a long time in a number of patients. Therefore, confirmation of olfactory dysfunction in early stage using objective modality and immediate initiation of proper management in COVID-19 patient with olfactory dysfunction is necessary.’

  1. A final point that I would deal with in the introduction or discussion is that related to the possible pathogenesis of persistent olfactory disturbances: persistence of the virus or of viral fragments in the olfactory epithelium? (Dias de Melo et al 10.1126/scitranslmed.abf8396) complete olfactory epithelium disruption? (vaira et al doi: 10.1017/S0022215120002455) local immune factors? (Saussez et al. doi: 10.1111/ene.14994)

Answer: Thank you for your comment. We added a paragraph about pathogenesis in introduction section as ‘Various studies have been reported on the pathogenesis of COVID-19 infection associated olfactory dysfunction. Kandemiril et al. published an article about olfactory bulb magnetic resonance imaging (MRI) finding in persistent COVID-19 anosmia patients. They reported that olfactory cleft opacification was observed in 73.9% of the patients, and 43.5% and 60.9% of the patients had below normal olfactory bulb and shallow olfactory sulci, respectively. In addition, 54.2% of the patients had the change of the normal shape (inverted J shape) of olfactory bulb, and 91.3% of the patients had abnormal signal intensity of olfactory bulb [12]. Moreover, Chiu et al reported a case about olfactory bulb atrophy after 2 months duration of COVID-19 infection associated olfactory dysfunction, compared with pre-COVID imaging [13]. Politi et al. also re-ported a case about serial evaluation of olfactory bulb using MRI during COVID-19 infection. They reported that olfactory bulbs were thinner and slightly less hyperintense at 28 days after onset of symptom. Therefore, the authors suggested that direct or in-direct injury to olfactory neuronal pathway lead to olfactory dysfunction in COVID-19 infection [14].

 According to a study about inflammatory cytokine evaluation in olfactory epithelium of the deceased patients due to COVID-19 infection by Torabi et al., they re-ported that level of TNF-α was significantly increased in the COVID-19 group than control group [15]. Therefore, they suggested that direct inflammation of the olfactory epithelium can lead to sensorineural olfactory dysfunction in COVID-19. Damage of olfactory epithelium is also confirmed by other studies. Li et al. reported that SARS-CoV-2 infects the ciliated cells in the human nasal epithelium, and cause deciliation. They suggested that this defect in olfactory cilia lead to olfactory loss after COVID-19 infection [16]. Vaira et al. also reported significant deterioration of olfactory epithelium using histopathological evaluation on the olfactory epithelium of COVID-19 patients who presented with anosmia more than 3 months of duration. In this study, the subjects also conducted contrast enhanced MRI of nasal cavity and brain, and found no abnormalities of the volume of olfactory bulb and cleft and no signal abnormality [17]. Therefore, they suggested that disruption and desquamation of the olfactory epithelium is the underlying mechanism in COVID-19 related olfactory dysfunction and these findings have important implications when considering treatment for olfactory dysfunction in COVID-19 patients, it should be considered that olfactory epithelium should be the target.

Recently, an interesting article was published about COVID-19 related anosmia is associated with viral persistence in human olfactory epithelium. They reported that in the patients with long-lasting/relapsing olfactory dysfunction after COVID-19 infection, SARS-CoV-2 RNA was detected in cytological samples from olfactory mucosa, but not in nasopharyngeal sample. Therefore, they regarded that persistence of COVID-19 associated olfactory dysfunction is linked to the inflammation caused by persistent infection [18]. In addition, they also suggested that persistent olfactory dysfunction might result from direct damage to the olfactory sensory neurons of olfactory epithelium and retrograde neuro-invasion of SARS-CoV-2 through olfactory route induced neuro-inflammation.’

Reviewer 2 Report

The systemic review covers Covid-19 OGD treatments quite comprehensively. Unfortunately, it does not follow standard EBM paper guidelines to make their arguments, especially, there is no quality check, no meta-analysis of the reviewed articles. Most of the summaries are descriptive and qualitative. 

Author Response

Re: jpm-1365638

Treatment and prognosis of COVID-19 associated olfactory and gustatory dysfunctions

Referee(s) comments to author: 

Reviewer: 2

Comments and Suggestions for Author

The systemic review covers Covid-19 OGD treatments quite comprehensively. Unfortunately, it does not follow standard EBM paper guidelines to make their arguments, especially, there is no quality check, no meta-analysis of the reviewed articles. Most of the summaries are descriptive and qualitative.

Answer: Thank you for your comment. In this article, our authors did not conduct the systematic review and meta-analysis. We only performed the literature review about the treatment and prognosis of COVID-19 associated olfactory and gustatory dysfunctions. Our authors also agree that the results presented through systematic review and meta-analysis suggest more valuable than literature review. Recently, many patients have complained of olfactory dysfunction after COVID-19 infection, but there is no validated treatment for it. Therefore, this study was conducted with the belief that it was necessary to summarize the treatment outcome by a physician who had experience in treating patients. However, for the systematic review of research on this topic, among the various clinical studies, the modality of olfactory outcome evaluation according to treatment was not made uniformly, and the clinical characteristics of the studies subjects showed difference. Therefore, we regarded that literature review is more suitable in the current situation. Because we conducted a literature review, based on the results of recently published several studies, we could only suggest some direction in the treatment of the current situation, but could not present a clear conclusion. As the COVID-19 pandemic continues, the number of patients complaining of olfactory dysfunction will continue to increase, and the demand for treatment will also increase. Therefore, if we summarize the current treatment outcome and share it, I think it will be helpful to many physicians for establish of treatment policy. In addition, if a certain amount of treatment is established and the results of various well-designed studies are published, valuable conclusions using systematic review and meta-analysis can be presented.

Round 2

Reviewer 1 Report

The authors responded to all my comments.

Reviewer 2 Report

The revision has not modified the points mentioned in previous comments.